# Protective Effects of Baicalin on Peritoneal Tight Junctions in Piglets Challenged with *Glaesserella parasuis*

**DOI:** 10.3390/molecules26051268

**Published:** 2021-02-26

**Authors:** Jiacheng Zhang, Zhaoran Zhang, Jianfeng Xu, Chun Ye, Shulin Fu, Chien-An Andy Hu, Yinsheng Qiu, Yu Liu

**Affiliations:** 1Hubei Key Laboratory of Animal Nutrition and Feed Science, School of Animal Science and Nutritional Engineering, Wuhan Polytechnic University, Wuhan 430023, China; zhangjiacheng0826@163.com (J.Z.); 15893715001@163.com (Z.Z.); xujianfeng2017@163.com (J.X.); yechun@whpu.edu.cn (C.Y.); shulinfu@whpu.edu.cn (S.F.); 2Biochemistry and Molecular Biology, University of New Mexico School of Medicine, Albuquerque, NM 87131, USA; AHU@salud.unm.edu

**Keywords:** baicalin, *Glaesserella parasuis*, tight junctions, peritoneum, piglets

## Abstract

*Glaesserella parasuis* (*G. parasuis*) causes inflammation and damage to piglets. Whether polyserositis caused by *G. parasuis* is due to tight junctions damage and the protective effect of baicalin on it have not been examined. Therefore, this study aims to investigate the effects of baicalin on peritoneal tight junctions of piglets challenged with *G. parasuis* and its underlying molecular mechanisms. Piglets were challenged with *G. parasuis* and treated with or without baicalin. RT-PCR was performed to examine the expression of peritoneal tight junctions genes. Immunofluorescence was carried out to detect the distribution patterns of tight junctions proteins. Western blot assays were carried out to determine the involved signaling pathways. Our data showed that *G. parasuis* infection can down-regulate the tight junctions expression and disrupt the distribution of tight junctions proteins. Baicalin can alleviate the down-regulation of tight junctions mRNA in peritoneum, prevent the abnormalities and maintain the continuous organization of tight junctions. Our results provide novel evidence to support that baicalin has the capacity to protect peritoneal tight junctions from *G. parasuis*-induced inflammation. The protective mechanisms of baicalin could be associated with inhibition of the activation of PKC and MLCK/MLC signaling pathway. Taken together, these data demonstrated that baicalin is a promising natural agent for the prevention and treatment of *G. parasuis* infection.

## 1. Introduction

*Glaesserella parasuis* (*G. parasuis*), previously named *Haemophilus parasuis*, is a Gram-negative bacterium and one of the most important bacteria affecting pigs as an early commensal colonizer in the upper respiratory tract of weaning pigs [1]. The disease caused by this pathogen is characterized by polyserositis and it is known as Glässer’s disease [2]. It can result in high mortality and morbidity, with significant economic losses for pig producers [3]. Because of the incomplete efficacy of current vaccines, antimicrobials are commonly used to treat *G. parasuis* infections [4]. However, the phenomenon of bacterial resistance has become more and more serious. Therefore, exploring the pathogenesis and finding alternative feasible ways of preventing and controlling *G. parasuis* infections has become very urgent.

Peritonitis, a common clinical symptom of *G. parasuis* infections, might stem from damage to peritoneal tight junctions [5]. The peritoneum is a membranous tissue mainly composed of mesothelial cells, which has a coating effect on most organs in the abdominal cavity, and can secrete mucus to alleviate the friction between organs [6]. The molecular correlation between the paracellular channel and this barrier function have been discovered, namely tight junctions, which show a tissue-specific expression of tight junction proteins determining the functional properties of the tissues [7]. Tight junctions are complex structure of different proteins, including integral membrane proteins (claudins, occludin, junctional adhesion molecules “JAMs”) and peripheral membrane proteins (zonula occludins or ZOs, such as ZO-1, ZO-2, and ZO-3) [8]. The stability of its function requires the coordination of multiple proteins [9]. An indispensable role of tight junctions involved in pathogen infection has been widely demonstrated since disruption of tight junctions leads to a distinct increase in paracellular permeability and polarity defects which facilitate viral or bacterial entry and spread [10]. Changes in the peritoneal mesothelial cell phenotype, including loss of tight junctions, may allow ectopic cells to bind to, or early lesions invade into, the extracellular matrix. Signaling pathways involved in the assembly, disassembly, and maintenance of tight junctions are controlled by a number of signaling molecules, such as protein kinase C (PKC), mitogen-activated protein kinases (MAPK), myosin light chain kinase (MLCK), and myosion light chain 2 (MLC-2) [11,12,13,14].

Many natural compounds such as flavonoids have been demonstrated to exhibit a broad spectrum of biological activities such as anti–inflammatory properties [15,16]. The available reports reveal that flavonoids such as baicalin, naringin, and hesperidin (Figure 1), have promotive and protective effects on tight junctions barrier functions [17,18,19,20]. Baicalin (7-glucuronic acid-5,6-dihydroxy-flavone), is one of the primary bioactive flavonoid compounds extracted from the roots of *Scutellaria baicalensis* Georgi. Baicalin is used clinically in humans and animals because of its antimicrobial, anti-inflammatory, antitumor and antioxidant properties [21,22,23,24]. Our previously work demonstrated that baicalin has anti–inflammatory effects in *G. parasuis*-challenged piglets in vivo and in vitro by suppressing inflammatory cytokines and HMGB1 via NF-κB and NLRP3 signaling and reversing apoptosis by altering PKC–MAPK signaling cells [25,26,27,28]. Baicalin is prominent in the literature on protection of tight junction in epithelial and endothelial cells [17,29,30], although high doses of baicalin can reduce tight junction integrity by partly targeting the first PDZ domain of ZO-1 [31]. However, whether an appropriate dose of baicalin can protect tight junctions to inhibit inflammation in piglets against *G. parasuis* challenge has not been investigated.

In this study, we attempted to investigate the protection effects of baicalin on tight junctions in the peritoneum of piglets challenged with *G. parasuis* and its underlying molecular mechanisms.

## 2. Results

### 2.1. Effects of Baicalin on the Expression of Tight Junctions Genes in Peritoneum of G. parasuis Challenged Piglets

Figure 2 shows the regulation of tight junction genes in peritoneum in all groups. Challenged with *G. parasuis* resulted in a significant decrease in the mRNA expressions of occludin, ZO-1, claudin-1, and JAM-1, compared with control group (*p* < 0.01). Ethyl pyruvate (EP) and flunixin meglumine (FM) were used as positive controls in the experiments. EP and FM could significantly up-regulate the gene expressions of ZO-1 and JAM-1 in piglets’ peritoneum (*p* < 0.01). FM could significantly up-regulate the gene expression of occludin in peritoneum (*p* < 0.01). EP could significantly up-regulate the gene expression of blaudin-1 (*p* < 0.01). Baicalin at 25, 50 and 100 mg/kg could significantly restore the expressions of occludin and ZO-1 in the peritoneum of piglets infected with *G. parasuis* (*p* < 0.01, *p* < 0.05). Baicalin at 50 mg/kg could significantly up-regulate the gene expression of claudin-1 (*p* < 0.01). Baicalin at 100 mg/kg could significantly up-regulate the gene expression of JAM-1 (*p* < 0.05).

### 2.2. Effect of Baicalin on the Distibution Patterns of Tight Junctions in Peritoneum of G. parasuis Challenged Piglets

In addition to appropriate expression, proper organization and distribution of the tight junction proteins is critical for the maintenance of a permeability barrier. The *G. parasuis* mediated disruption of the distribution results of tight junctions proteins in peritoneum and the effect of each drug are shown in Figure 3, Figure 4, Figure 5 and Figure 6. As expected, *G. parasuis* infection alters the distribution of occludin, ZO-1, claudin-1 and JAM-1 in the peritoneum. We observed that occludin, ZO-1, claudin-1 and JAM-1 protein staining appeared to be reduced and more fragmented in the *G. parasuis* group. Administration with EP or FM could significantly prevent these abnormalities and maintain the continuous organization of tight junctions (*p* < 0.01). Treatment with 25, 50 and 100 mg/kg baicalin could significantly attenuate this disorganization (*p* < 0.01). Among all drug treatment groups, 50 mg/kg baicalin was demonstrated the best protection effect on the disruption of the distribution of tight junction proteins. These findings identify a *G. parasuis*-induced global impairment of tight junction integrity, a process largely prevented by baicalin supplementation.

### 2.3. Effect of Baicalin on PKC and MLCK/MLC Signaling Pathways in Peritoneum of G. parasuis Infected Piglets

Western blot assays were carried out to determine whether the protective mechanism of baicalin on tight junctions of *G. parasuis*-infected piglets acts through the PKC and MLCK/MLC pathways. As shown in Figure 7a,b, *G. parasuis* infection significantly increased the phosphorylation level of PKC-α (*p* < 0.01), while the protein level of PKC-α remained unchanged (*p* > 0.05). EP, FM, and baicalin could down-regulate the expression of p-PKC-α induced by *G. parasuis* and have no effect on the expression of PKC-α.

Compared to control group, the MLCK protein was significantly increased in the peritoneum in *G. parasuis* group (*p* < 0.01) (Figure 7c). EP and FM could significantly inhibit the expression of MLCK protein. Baicalin at 25, 50 and 100 mg/kg significantly altered the increasing effect of *G. parasuis* on the expression of MLCK (*p* < 0.01) (Figure 7c).

*G. parasuis* infection markedly elevated the phosphorylation level of MLC-2 (*p* < 0.01), yet the protein level of MLC-2 remained almost unified between the control and *G. parasuis* group (*p* > 0.05). Administration of EP, FM and baicalin could significantly down-regulated the expression of p-MLC-2 and have no effect on the expression of MLC-2 (Figure 7d,e).

### 2.4. Histopathological Analysis

Many fibrotic exudates and abdominal organs adhesion were observed in the abdominal cavity in the *G. parasuis* group piglets. Fibrotic lesions were found in peritoneum in *G. parasuis* group. The histopathological analysis was performed to estimate the extent of damage of peritoneum tissues. As shown in Figure 8a, the result of histopathologic analysis displayed no histopathologic changes in control group. The piglets from the *G. parasuis* infection group displayed severe pathological damage as inflammatory cell infiltration and aggregation to their peritoneum tissues (Figure 8b). Moderate inflammatory cell infiltration and aggregation was found in peritoneum in EP treatment group (Figure 8c). Only mild tissue damage was detected in the surviving piglets of the FM treatment group (Figure 8d). The inflammatory cells infiltration was reduced, and the structure of the peritoneum was comparatively complete in the baicalin treatment groups.

## 3. Discussion

*G. parasuis* is the main pathogen of Glässer’s disease, characterized by polyserositis [2]. The fibrotic inflammation caused by *G. parasuis* mainly occurs in serosa. It is originated from tight junction destruction and serosa damage, which triggers a damaged barrier function and eventually leads to fibrotic exudation [32]. Autopsy results after *G. parasuis* infection show that the organs of piglets are seriously adhered and there are fibrotic exudates on the surface of the pleura and peritoneum. A large number of inflammatory cells are infiltrated into the peritoneum, containing a small number of necrotic cells, which were filamentous, which is consistent with the characteristic lesions of *G. parasuis*. EP and FM were used as positive control drugs in the experiment derived from their anti-inflammatory effects to control inflammation-related diseases, such as *G. parasuis* infection [28]. Under the action of baicalin, the peritoneum lesions were alleviated to a certain degree, which showed the protective effects of baicalin on the damage of peritoneal tight junctions structures in piglets infected with *G. parasuis*, and provided the basis for exploring its mechanism of action on tight junctions.

Tight junctions are closed complexes at the top of the lateral membrane interface of adjacent epithelial and mesothelial cells. Occludin, claudin and JAMs form a tight junctions skeleton. ZOs are the bridge between cytoskeleton and transmembrane proteins [33]. ZOs proteins interact directly with most transmembrane proteins located at tight junctions. Occludin is involved in the regulation of cell surface receptor signal transduction. In vivo and in vitro studies have shown that occludin is a key regulator of the tight junctions barrier, and multiple domains of occludin are involved in regulating cell bypass permeability [34]. Lack of occludin can cause moderate dysfunction of tight junctions or dysfunction of other cellular signaling pathways related to occludin [35]. Inflammatory injury can cause abnormal distribution, reduce the expression and dissolution of ZO-1 protein, damage the tight junctions structures between cells, widen the intercellular space, and increase the permeability of intestinal epithelial cells [36]. Studies have shown that LPS can reduce the content of ZO-1 in intestinal epithelial cells, and baicalin can protect the ZO-1 damage caused by LPS, which is consistent with the results of our experiment [17].

To our knowledge, this is the first study where tight junctions proteins alterations in *G. parasuis* infected piglets were explored. The mRNA expressions of occludin, ZO-1, claudin-1, and JAM-1 in peritoneum of *G. parasuis* infected piglets were significantly down-regulated. The distributions of each protein in the peritoneum of *G. parasuis*-infected piglets were significantly disrupted. Baicalin can attenuate the down-regulation of each mRNA in peritoneum, prevent these abnormalities and maintain the continuous organization of tight junctions. These results demonstrated that baicalin can alleviate peritoneal tight junction alterations caused by *G. parasuis* infection. The protective effects of baicalin on tight junctions are superior to EP and FM.

The PKC family is a phospholipid dependent serine/threonine protein kinase activated by calcium, which is widely distributed in the body and plays an important role in the regulation of tight junctions [12,37,38,39,40]. Activation of PKC will increase cell permeability, which is critical for tight junction regulation. Studies have shown that PKC mediates the phosphorylation of occludin and its regulation in cell distribution. PKC regulates the assembly of tight junction complex through the phosphorylation of connexin 43 and closure protein to coordinate the formation of functional active barrier and the function of intercellular channel [41]. PKC plays a key role in the translocation of ZO-1 from the cell interior to cell membranes, which can affect the regulation and formation of tight junctions. At the same time, ZO-1 is also the existence of cytoskeleton in PKC signal transduction pathway on cell membrane junction surface [42]. In the process of tight junction decomposition, ZO-2 is phosphorylated by the atypical PKC serine located at tight junctions and combined with ZO-2 [43,44]. The isotypes of PKC and ZO-1 are co-located on the side of plasma membrane, and ZO-1 may be directly phosphorylated by PKC during tight junction assembly [37].

In this study, the protein change of PKC-α in the peritoneum of piglets infected with *G. parasuis* was detected. Compared with the control group, the expression of phosphorylation of PKC-α in peritoneum of *G. parasuis*-infected piglets was significantly increased and no significant change was found in non-phosphorylated PKC-α, indicating that PKC–α in peritoneum was activated by *G. parasuis*. Baicalin can attenuate the phosphorylation of PKC-α, which in consistent with our previous work and the results of other studies of the regulation effects of baicalin on PKC [27,45,46,47]. The protection effect of baicalin on tight junction abnormalities may relate directly on inhibition of PKC and/or the downstream signaling pathways such as MLCK/MLC.

MLCK is a serine/threonine protein kinase that has an important role in the reorganization of the cytoskeleton leading to disruption of barrier integrity [48]. MLCK catalyzes the phosphorylation of MLC proteins to stimulate the contraction of actin/myosin peri-junctional filaments and consequent tight junctions permeabilization [49]. There is sufficient evidence that tight junctions proteins are regulated by MLC-2, which principally depends upon the activation of MLCK [13,14,50,51]. The increased MLCK is indicative of tight junction barrier disruption induced by pro-inflammatory cytokines. Our results showed that the MLCK and p-MLC-2 proteins were significantly increased in the peritoneum in *G. parasuis* challenged piglets, suggesting that *G. parasuis* infection could cause tight junctions barrier disruption. Baicalin can inhibit the increasing protein levels of MLCK and p-MLC-2 induced by *G. parasuis*. These results suggest that the protective effects of baicalin on tight junctions in peritoneum are derived from inhibiting MLCK/MLC pathway.

## 4. Materials and Methods

The study was carried out at Animal Experimental Base (Wuhan, Hubei, China) in Sinopharm Animal Health Corporation Ltd. All the experimental procedure and operations used in the management and care of piglets were in agreement with the Wuhan Polytechnic University Laboratory Animals Welfare and Animal Experimental Ethical Inspection (reference number WP20100501).

### 4.1. Bacterial Strains

*Glaesserella parasuis* strain SH0165 serovar 5 was used, which was isolated from the lung of a commercially produced pig with the typical characteristics of Glässer’s disease. The SH0165 isolate was cultured at 37 °C for 12 h in tryptic soy broth (Difco, Lawrence, KS, USA) or grown for 24 h in tryptic soy agar (Difco) supplemented with nicotinamide adenine dinucleotide (Sigma, St. Louis, MO, USA) and foetal bovine serum (Gibco, Gaithersburg, MD, USA).

### 4.2. Experimental Products

Baicalin was purchased from National Institutes for Food and Drug Control (B110715-201318, Beijing, China). Sodium baicalin was prepared at the Hubei Key Laboratory of Animal Nutrition and Feed Science (Wuhan, China) and was >95% pure [52]. Ethyl pyruvate (EP) and flunixin meglumine (FM) were purchased from Shanghai Macklin Biochemical Co., Ltd. (Shanghai, China) and Guangdong WenS Dahuanong Biotechnology Co., Ltd. (Yunfu, China), respectively.

### 4.3. Experimental Animals, Management, and Design

A total of 56 weaned healthy piglets (Duroc × Landrace × Large White, 23-d weaned) weighing 8 to 10 kg were purchased from Wuhan Wannianqing Animal Husbandry Co., Ltd. (Wuhan, China). The piglets were confirmed negative for antibodies directed against *G. parasuis* using INgezim Haemophilus 11.HPS.K.1 (Ingezim, Madrid, Spain).

The 56 piglets were randomly divided into seven groups, each group consisting of eight piglets. They were: control group, *G. parasuis* group, EP + *G. parasuis* group, FM + *G. parasuis* group, and 3 baicalin + *G. parasuis* groups (dose 25, 50 and 100 mg/kg b.w., respectively). Drug treatment and challenge: The piglets in EP + *G. parasuis* group were injected intraperitoneally with EP at 40 mg/kg b.w. The piglets in FM + *G. parasuis* group were injected intramuscularly with FM at 2 mg/kg b.w. In baicalin + *G. parasuis* groups, sodium baicalin dissolved in saline was administered intramuscularly at 25, 50, and 100 mg/kg b.w., respectively. After 30 min of the drug treatment, all the piglets except those in control group were challenged intraperitoneally with 1 × 10^9^ CFU of the *G. parasuis* strain (SH0165) in 2 mL of normal saline. The piglets in the control group were injected intraperitoneally with an equivalent volume of saline. Dosing of EP, FM, and baicalin were performed twice daily with an interval of 6 h until the day of post-mortem examination.

### 4.4. Experimental Sample Collection

On the 8th day after *G. parasuis* challenge, the living piglets were humanely euthanized by intravenous injection of sodium pentobarbital, followed by exsanguination. Peritoneum tissue samples were collected, stored at −80°C for further experiment processing detection or fixed in 4% paraformaldehyde for pathological examination.

### 4.5. RNA Extraction and RT-PCR

The total RNA was isolated from peritoneum homogenates using RNA prep pure Cell/Bacteria Kit following the manufacturer’s instructions (Tiangen, Beijing, China). The RNA was reverse transcribed into cDNA using the Reverse Transcription Kit following the manufacturer’s instructions (Takara Biotechnology, Kusatsu, Japan). Specific expression primers for Claudin-1, JAM-1, ZO-1, Occludin and β-actin were designed using Primer 6.0 (Premier, West Toronto, ON, Canada). The primers used for RT-PCR are listed in Table 1. PCR was performed according to the following conditions: 95 °C for 5 min followed by 32 cycles of amplification at 95 °C for 30 s, Tm temperature for 32 s and 72 °C for 30 s, then a final extension at 72 °C for 5 min. The densities of each band were quantified using a gel imaging system (Tanon 4100, Tanon, Shanghai, China) and a ratio was calculated using β-actin as a control. The quantitative results for fluorescence were calculated by 2^−ΔΔCt^ using the normalization method.

### 4.6. Immunofluorescence Microscopy

Immunofluorescence was carried out to detect the distribution patterns of claudin-1, occludin, ZO-1 and JAM-1 proteins in the peritoneum. Thin sections (3 mm) of paraffin-embedded sections of peritoneum were prepared and mounted into adhesive microscopic glass slides. After dewaxing, the sections were permeabilized with citrate buffer for 15 min in microwave, washed 3 times with phosphate buffered saline (PBS) and then blocked with 5% GSA (diluted in PBS) for 1.0 h at RT. The sections were incubated overnight with rabbit anti-occludin, anti-ZO-1, anti-claudin-1, and anti-JAM-1 antibody, respectively, at a 1:100 dilution at 4 °C, then incubated with Cy3-labeled goat anti rabbit (Beyotime, Shanghai, China) at a dilution of 1:100 for 1 h at RT. Fluorescence images were captured using a confocal microscope. The Image J software 1.8.0 (National Institutes of Health, Bethesda, MD, USA) was used to evaluate the amounts of each protein present at the intercellular junctions by semi-quantitatively measuring fluorescence density in the selected areas.

### 4.7. Western Blotting Analysis

The peritoneum samples of the piglets were collected, dissociated by RIPA lysis buffer supplemented with protease inhibitor mixture and centrifuged at 12,000 *g* for 15 min at 4 °C. The total protein was measured with BCA protein extraction kit (Beyotime). Subsequently, samples with the same amount of protein (80 μg) were fractionated using 10% SDS-PAGE and then transferred onto polyvinylidene fluoride (PVDF) membranes. After blocked with 5% skimmed milk for 3 h, the PVDF membranes were incubated with special primary antibody (containing 5% BSA TBS-T solution, 1:1000 dilution, rabbit anti-β-actin, anti-PKC-α, anti-p-PKC-α, Cell Signaling, Danvers, MA, USA; anti-MLCK, anti-MLC-2 and anti-p-MLC-2, Abcam, Shanghai, China) at 4 °C overnight, and then incubated with the corresponding HRP labeled secondary antibodies (1:4000 dilution) at 37 °C for 3 h. Protein level was determined using the enhanced chemiluminescent (ECL) reagent (Beyotime) and the images were captured with a ChemiDoc MP Imaging System (Bio-Rad, Hercules, CA, USA). Quantitative analysis was carried out using FluorChem FC2 (Alpha Innotech, San Leandro, CA, USA). The β-actin was used as the inner loading control. Gray value was analyzed and the relative expression level of protein was obtained.

### 4.8. Histopathology

Peritoneum histopathology was evaluated via haematoxylin and eosin (H&E) staining. Peritoneum were fixed in 10% neutral-buffered formalin, and embedded in paraffin. The sections (4 μm) were stained with H&E with the standard method and observed with a light microscope.

### 4.9. Statistical Analysis

SPSS Statistics version 17.0 (IBM, Armonk, NY, USA) was used for statistical analysis. The data were shown as the mean ± standard deviation (SD). The differences between the data sets were assessed by one-way analysis of variance (ANOVA) and multiple comparisons between the groups were performed using LSD method. Probability value of *p* < 0.05 was considered significant.

## 5. Conclusions

In conclusion, our results provide novel evidence to support the notion that *G. parasuis* can downregulate peritoneal tight junction gene expressions and disrupt the distribution of tight junction proteins. Baicalin has the capacity to protect peritoneal tight junctions from *G. parasuis*-induced injuries. The protective mechanisms of baicalin could be associated with the inhibition of the activation of the PKC and MLCK/MLC signaling pathways. The pharmacological features of baicalin thus make it a promising natural antimicrobial compound for prevention and treating of Glässer’s disease.

## Figures and Tables

**Figure 1 molecules-26-01268-f001:**
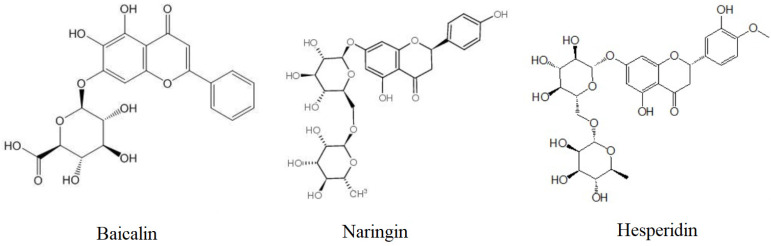
Chemical structures of flavonoids, baicalin, naringin, and hesperidin.

**Figure 2 molecules-26-01268-f002:**
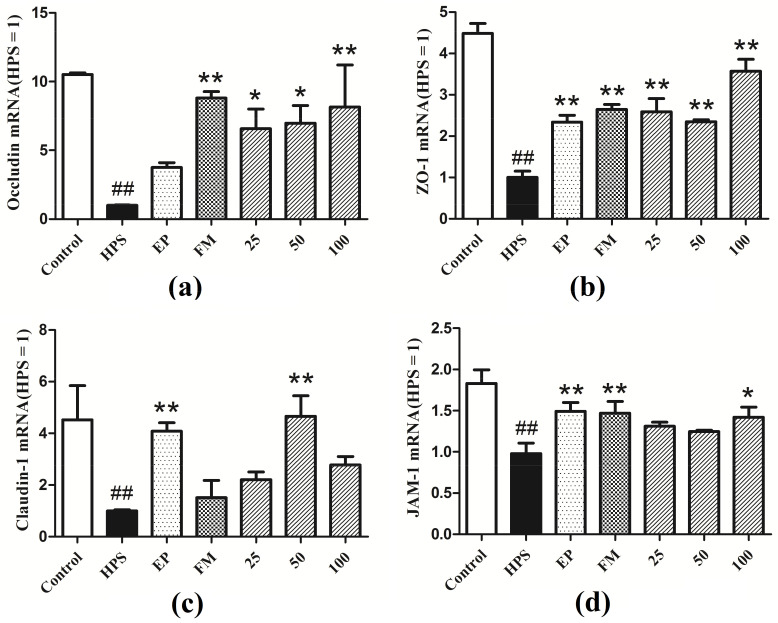
Expressions of tight junctions genes (**a**) Occludin, (**b**) ZO-1, (**c**) Claudin-1, (**d**) JAM-1 in the peritoneum of piglets challenged with *G. parasuis* (Mean ± SD, *n* = 3). HPS: *G. parasuis* group, EP:EP + *G. parasuis* group, FM:FM + *G. parasuis* group, 25:25 mg/kg baicalin + *G. parasuis* group, 50:50 mg/kg baicalin + *G. parasuis* group, 100:100 mg/kg baicalin + *G. parasuis* group. ^##^
*p* < 0.01 vs control. * *p* < 0.05 vs *G. parasuis* group, and ** *p* < 0.01 vs *G. parasuis* group.

**Figure 3 molecules-26-01268-f003:**
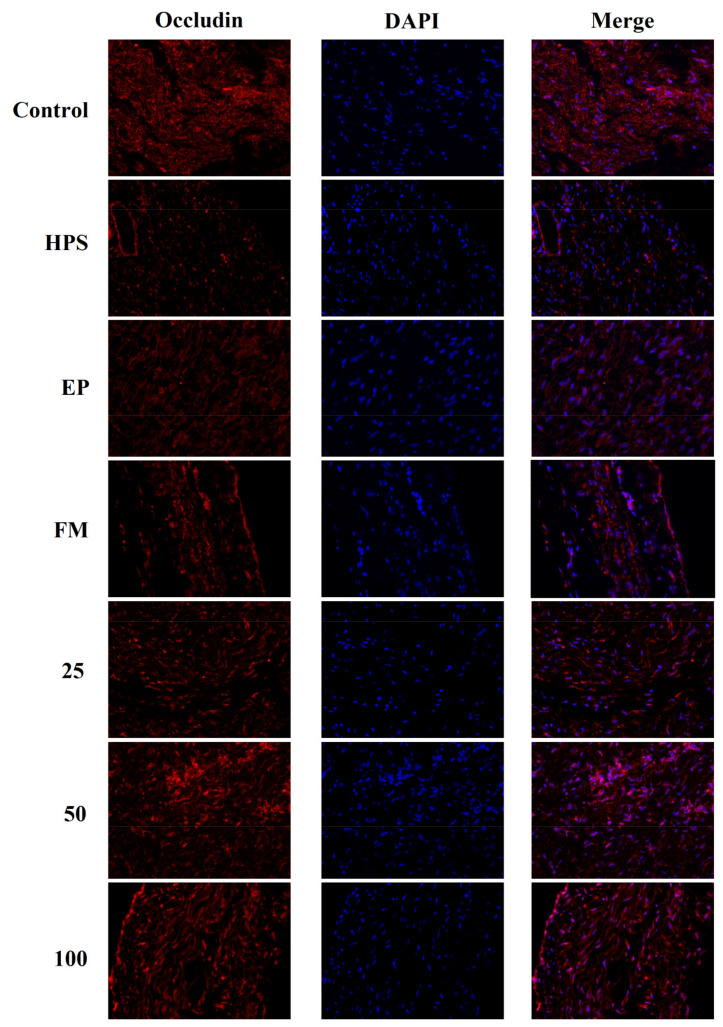
Effect of baicalin on the distribution patterns of Occludin in the peritoneum of piglets challenged with *G. parasuis* (magnification 10 × 40). HPS: *G. parasuis* group, EP:EP + *G. parasuis* group, FM:FM + *G. parasuis* group, 25:25 mg/kg baicalin + *G. parasuis* group, 50:50 mg/kg baicalin + *G. parasuis* group, 100:100 mg/kg baicalin + *G. parasuis* group.

**Figure 4 molecules-26-01268-f004:**
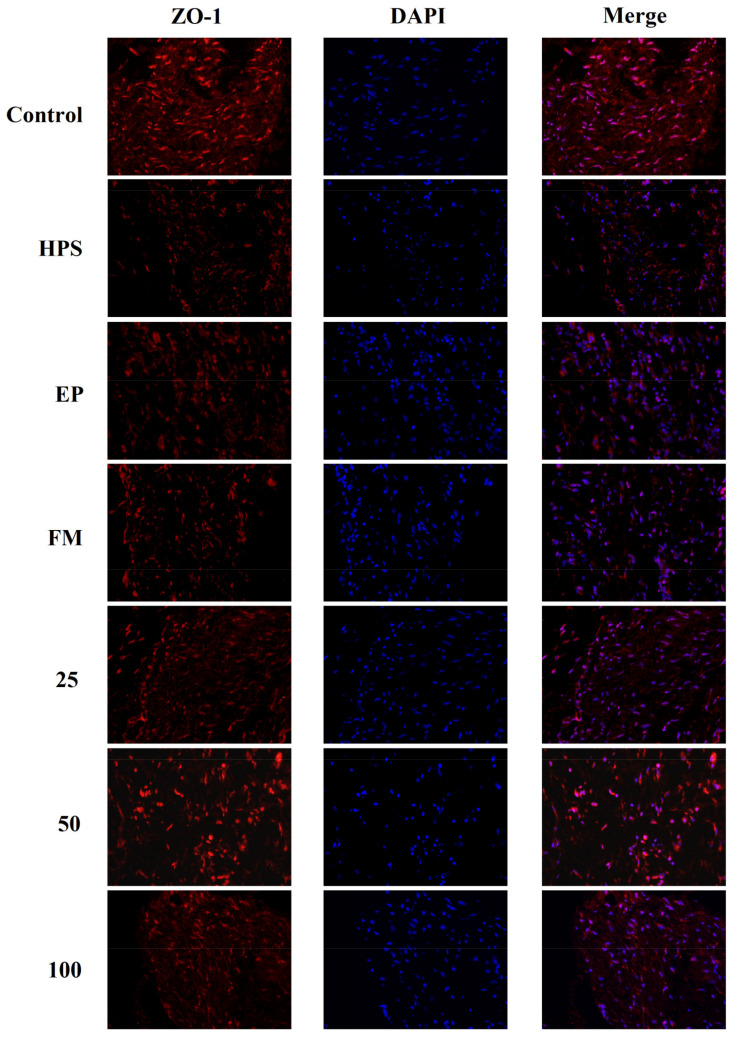
Effect of baicalin on the distribution patterns of ZO-1 in the peritoneum of piglets challenged with *G. parasuis* (magnification 10 × 40). HPS: *G. parasuis* group, EP:EP + *G. parasuis* group, FM:FM + *G. parasuis* group, 25:25 mg/kg baicalin + *G. parasuis* group, 50:50 mg/kg baicalin + *G. parasuis* group, 100:100 mg/kg baicalin + *G. parasuis* group.

**Figure 5 molecules-26-01268-f005:**
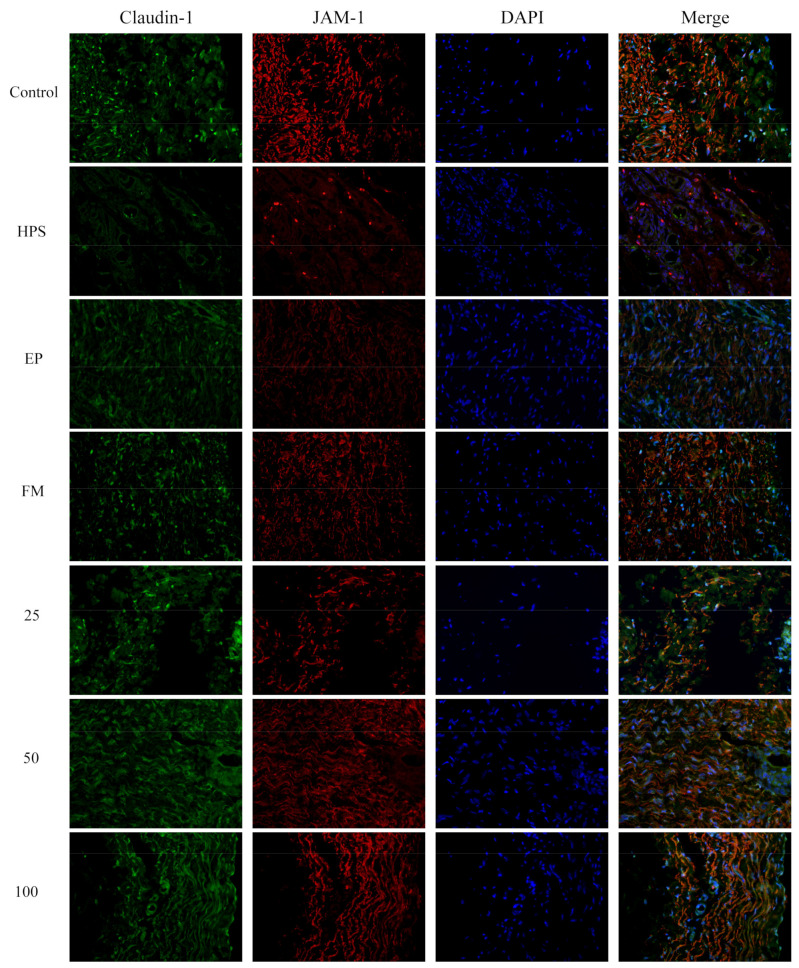
Effect of baicalin on the distribution patterns of claudin-1 and JAM-1 in the peritoneum of piglets challenged with *G. parasuis* (magnification 10 × 40). HPS: *G. parasuis* group, EP:EP + *G. parasuis* group, FM:FM + *G. parasuis* group, 25:25 mg/kg baicalin + *G. parasuis* group, 50:50 mg/kg baicalin + *G. parasuis* group, 100:100 mg/kg baicalin + *G. parasuis* group.

**Figure 6 molecules-26-01268-f006:**
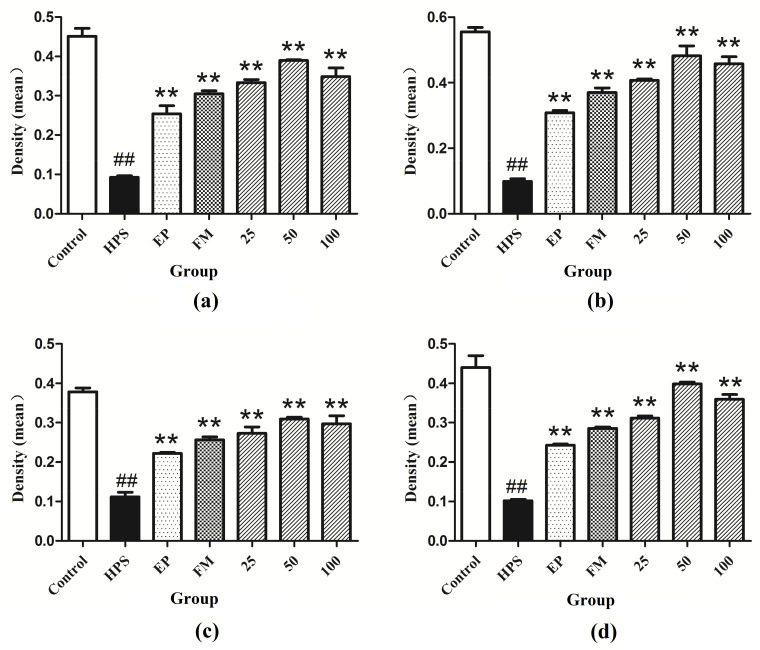
Fluorescence of (**a**) occludin, (**b**) ZO-1, (**c**) claudin-1 and (**d**) JAM-1 in peritoneum were measured by densitometric analysis using the software Image J. HPS: *G. parasuis* group, EP:EP + *G. parasuis* group, FM:FM + *G. parasuis* group, 25:25 mg/kg baicalin + *G. parasuis* group, 50:50 mg/kg baicalin + *G. parasuis* group, 100:100 mg/kg baicalin + *G. parasuis* group. ^##^
*p* < 0.01 vs. control. and ** *p* < 0.01 vs. *G. parasuis* group.

**Figure 7 molecules-26-01268-f007:**
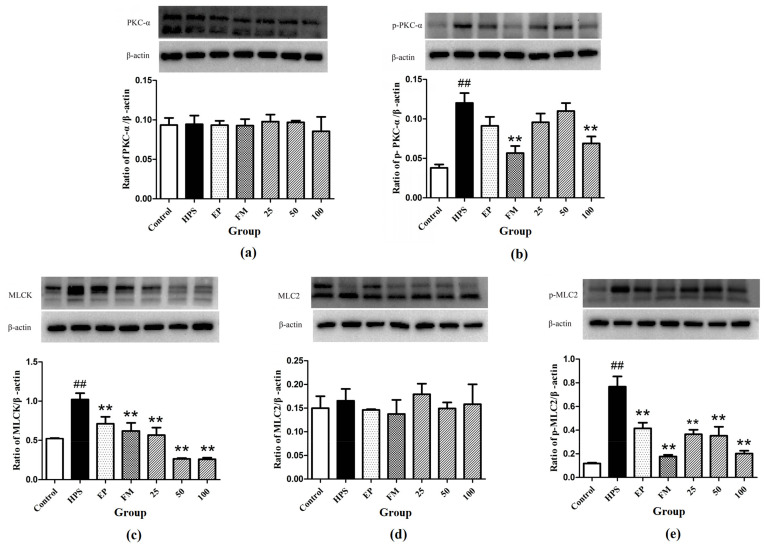
Inhibition effects of baicalin on PKC and MLCK/MLC signaling pathway in peritoneum activated by *G. parasuis* (**a**) PKC-α, (**b**) p-PKC-α, (**c**) MLCK, (**d**) MLC-2, (**e**) p-MLC-2 (Mean ± SD, *n* = 3). HPS: *G. parasuis* group, EP:EP + *G. parasuis* group, FM:FM + *G. parasuis* group, 25:25 mg/kg baicalin + *G. parasuis* group, 50:50 mg/kg baicalin + *G. parasuis* group, 100:100 mg/kg baicalin + *G. parasuis* group. ^##^
*p* < 0.01 vs. control and ** *p* < 0.01 vs. *G. parasuis* group.

**Figure 8 molecules-26-01268-f008:**
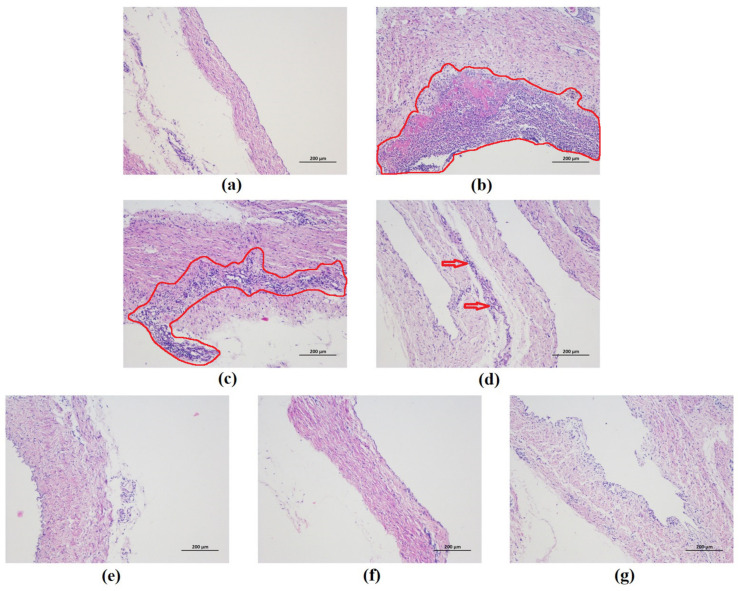
Histopathological change in the piglet peritoneum after *G. parasuis* infection (H&E, ×100). (**a**) control group, (**b**) *G. parasuis* group, (**c**) EP + *G. parasuis* group, (**d**) FM + *G. parasuis* group, (**e**) 25 mg/kg baicalin + *G. parasuis* group, (**f**) 50 mg/kg baicalin + *G. parasuis* group, (**g**) 100 mg/kg baicalin + *G. parasuis* group. Severe and moderate lesions were circled with red line. Mild lesions were pointed by red arrow.

**Table 1 molecules-26-01268-t001:** Primer sequences for Q-RT PCR.

Gene	Nucleotide Sequences (5’–3’)		T_m_ (°C)	Length (bp)
β-actin	Forward	TGCGGGACATCAAGGAGAAG	57.4	216
	Reverse	AGTTGAAGGTGGTCTCGTGG	57.4	
Claudin-1	Forward	CCTTGCTGAATCTGAACAC	49.5	135
	Reverse	GCACCTCATCATCTTCCAT	50.0	
JAM-1	Forward	TGACAGAACAGGCGAATG	50.1	167
	Reverse	GCAGCATAGGCAGGAATT	50.1	
ZO-1	Forward	GAAGATGATGAAGATGAGGATG	50.3	184
	Reverse	GGAGGATGCTGTTGTCTC	49.9	
Occludin	Forward	GAGTGATTCGGATTCTGTCT	50.3	181
	Reverse	TAGCCATAACCATAGCCATAG	50.2	

## Data Availability

All relevant data have been presented as an integral part of this manuscript.

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
