# Peer review of "Protective Effects of Baicalin on Peritoneal Tight Junctions in Piglets Challenged with Glaesserella parasuis"

_molecules, 2021, doi:10.3390/molecules26051268_

Round 1
Reviewer 1 Report
The authors preented an in vivo study of the beneficial effects of baicalin on the peritoneal thight junctions in piglets challenged with G. parasuis.
line 61: change to "compounds extracted"
line 71: correct "chanllenged"
line 74: revise heading of paragraph 2.1
line 76: correct "each tight junction genes"
text in general: P or p for the statistics?
line 95: change to "are critical"
line 97: change to "each drug are shown in Figures 2-5"
line 100: please italicize "G. parasuis" and correct "administrtion"
line 132: change to "acts through"
paragraph 3 page 9: cellulosic? cellulose (twice)?
line 218: change to "The isotypes of"
line 282: euthanized with what?
Reviewer 2 Report
The effects of baicalin on tight junctions are well known. The novelty of the manuscript comes from the specific in vivo model used, piglets infected with G. parasuis.
The introduction should include a discussion on the effects of similar flavonoids and their aglycones on tight junctions. For example, papers by Hisada and the first showed baicalin reduced tight junction integrity
https://doi.org/10.1016/j.ejphar.2020.173436
https://doi.org/10.3390/nu12113285
The structure of baicalin and other natural products with this activity should be given in a figure.
Reviewer 3 Report
This manuscript presents the effects of the flavonoid baicalin on peritoneal tight junctions of piglets infected with the pathogen Glaesserella parasuis.
There are several shortcomings in the manuscript.
- What was the motivation to examine baicalin and not other flavonoids or other phytochemicals in general?
- The introduction is lacking some background information. What is currently used to control this disease? Vaccination? Antibiotics? Anti-inflammatory drugs?
- Line 61: In addition to “Scutellariae radix” it would be better to provide the Linnean binomial. Is this one species of Scutellaria or several?
- Lines 68-69: This seems to be an investigation with limited scope. Why only baicalin? There are numerous flavonoids with anti-inflammatory activity (see, for example: Pan MH, Lai CS, Ho CT. Anti-inflammatory activity of natural dietary flavonoids. Food & function. 2010;1(1):15-31.).
- Results section, line 79: “EP” and “FM” should be defined. Presumably these are positive controls, but it should be pointed out.
- Figures 3 and 4 are too small to see. Since these are graphically represented in Figure 5 they can probably be omitted.
- Figure 7: It is not clear what we are looking at. Can the authors provide some pointers and annotations on each photograph to help the reader?
- Line 168: “Glässer’s disease” rather than “Glazer's disease”.
- The conclusion that “The pharmacological features of baicalin make it a promising natural antimicrobial compound for prevention and treating of Glässer’s disease” is weak considering the relative costs (Sigma-Aldrich) of baicalin ($5, $10, $20 / kg bw) compared to the positive controls EP (ca. $0.04/kg bw), FM (ca. $2/kg bw).
Round 2
Reviewer 3 Report
The authors have adequately addressed the concerns of this reviewer and appropriate corrections and additions to the manuscript have been carried out. Publication without additional revision is recommended.
This manuscript is a resubmission of an earlier submission. The following is a list of the peer review reports and author responses from that submission.